# Significance of Smoking in Patients with Acute ST Elevation Myocardial Infarction (STEMI) Undergoing Primary Percutaneous Coronary Intervention: Evaluation of Coronary Flow, Microcirculation and Left Ventricular Systolic Function

Mariana Boulos [1,2], Yasmine Sharif [3], Nimer Assy [1,2] and Dawod Sharif [3,4,*]

1   Internal Medicine Department A, Galilee Medical Center, Nahariya 221001, Israel; marianab@gmc.gov.il (M.B.); nimera@gmc.gov.il (N.A.)
2   The Azrieli Faculty of Medicine, Bar-Ilan University, Safed 1311502, Israel
3   Technion, Israel Institute of Technology, Haifa 3200003, Israel
4   Department of Cardiology, Bnai Zion Medical Center, Haifa 3339419, Israel
*   Correspondence: dawod.sharif@b-zion.org.il; Tel.: +972-506-267277

**Abstract:** In the thrombolytic care era, myocardial infarction in cigarette smokers was associated with better six-month outcomes compared to non-smokers. Aims: We tested the hypothesis that in patients with anterior myocardial infarction with ST-segment elevation (STEMI) treated with primary percutaneous coronary intervention (PPCI), cigarette smoking is associated with better coronary artery flow, myocardial perfusion, and left ventricular systolic function. Methods: Ninety-nine patients (sixty-six smokers) with anterior STEMI treated with PPCI were studied. Angiographic coronary artery flow TIMI grades, myocardial blush grades (MBGs) before and after PPCI, ST-segment elevation resolution, maximal troponin I and creatine phosphokinase blood levels, left ventricular echocardiographic systolic function as well as left anterior descending coronary artery (LAD) velocity parameters at admission and at discharge were evaluated. Results: Smokers and non-smokers were treated similarly. In smokers, the age was significantly younger, $54 \pm 10$, compared to non-smokers, $71.8 \pm 10$ years, $p < 0.05$, and had a lower prevalence of women, 13.6% compared to 36.6%. TIMI and MBG before and after PPCI were similar between smokers and non-smokers. Smokers had a lower prevalence of complete ST elevation resolution, 33% compared to 50% in non-smokers. Diastolic LAD velocity and integral were lower in smokers, $p < 0.05$. Maximal biomarker blood levels as well as LV systolic function at admission and on discharge were similar. Conclusions: Cigarette smokers with anterior STEMI treated with PPCI were younger with a lower prevalence of women and of complete ST elevation resolution and had lower LAD diastolic velocity and integral late after PPCI. However, angiographic parameters and LV systolic function parameters were similar.

**Keywords:** smoking; anterior STEMI; primary coronary angioplasty

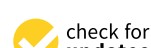



## 1. Introduction

Many epidemiological studies have shown that cigarette smoking is accompanied by a higher incidence of myocardial infarctions and death from coronary artery disease [1–4]. However, a previous study [5] has shown that thrombolytic treatment in smoking patients with myocardial infarctions was accompanied by much better results during hospitalization and after six months compared to non-smoking patients. Today, the treatment of choice for acute ST elevation myocardial infraction (STEMI) is primary percutaneous coronary intervention (PPCI) [6–9]. The goal of PPCI is to restore coronary flow to the heart muscle fed by the blocked artery. To maintain the artery's patency after PPCI, patients must be given a combined antiplatelet treatment including aspirin and an adenosine diphosphate receptor blocker. Such a combination in patients with acute coronary syndrome reduces

mortality and morbidity, and it was common to treat patients with clopidogrel [10]. Smoking increases the level of the active metabolite of clopidogrel which can benefit the patients. In an analysis of the studies' subgroups, the question arose as to whether clopidogrel only helps smokers [11]. Clopidogrel is a prodrug which requires two metabolic steps of activation to reach the active metabolite. Cigarette smoking increases the activity of cytochrome P450 isoenzyme 1A2 (CYP1A2) which is important in the first phase of clopidogrel's metabolism. It has been estimated that smoking increases the availability of the active metabolite of clopidogrel which increases its activity in smokers. A meta-analysis of several studies showed that the decrease in cardiovascular mortality, myocardial infarction and cerebrovascular events occurs mainly in smokers and very little in non-smokers [12–15].

After PPCI, the goal is to return the perfusion to the myocardium, distal to the blockage in the coronary artery, and thus bring about the recovery of the left ventricular function which was damaged during the blockage of the artery. The evaluation of the flow in the coronary artery after PPCI is carried out using the thrombolysis in myocardial infarction (TIMI) classification [16–19] and the flow evaluation and microcirculation function is carried out using myocardial blush grades (MBGs). After PPCI in patients with acute STEMI, damage to the microcirculation (microvascular injury) is the most common reason for a decrease in microcirculatory perfusion. The function of the coronary microcirculation influences the recovery of the left ventricular function after PPCI. There are several factors that are thought to cause myocardial perfusion impairment, including the microemboli of platelets, white blood cells, and ischemic necrosis [20,21].

Evaluating the coronary and myocardial flow in the catheterization room is important, but these evaluations are semi-quantitative and invasive and therefore difficult to repeat if necessary. New studies have shown that by means of a transthoracic Doppler echocardiography, it is possible to sample the blood flow in the coronary artery [22–25], and it is also possible to assess the function of the coronary microcirculation via the diastolic deceleration time (DDT), which is derived from the diastolic blood velocity curve in the artery. A DDT greater than 600 ms was found to be optimal, while a DDT less than 600 ms or systolic flow reversal are suboptimal and associated with a lack of recovery of left ventricular function after PPCI. It was also found that the microcirculation function is dynamic and changes after PPCI [26,27]. Therefore, in the current study, a transthoracic Doppler echocardiogram was used as an additional test to evaluate the flow and coronary microcirculation.

### 1.1. The Study Hypothesis

The study hypothesis is that coronary flow, myocardial flow, and left ventricular function in patients with acute anterior infarction and ST segment elevation undergoing primary percutaneous coronary intervention are better in smoking patients receiving clopidogrel compared to nonsmoking patients.

### 1.2. The Research Importance

If it turns out that cigarette smoking has a beneficial effect on the outcome of PPCI in patients with acute STEMI who received treatment with clopidogrel, there would be reason to still consider treatment with this drug in at least some of these patients.

## 2. Materials and Methods

This is an analytical study. The relevant data regarding the patients were received and recorded in the medical files prospectively, but their use and analysis was carried out retrospectively (Figure 1).

Ninety-nine patients with ST elevation acute anterior infarction who underwent coronary catheterization and primary PPCI percutaneous coronary intervention, and the target artery was the left anterior descending (LAD) artery, were examined. Blood tests performed included counts and extensive chemistry and, in addition, the common heart muscle marker tests, which included troponin and CPK.

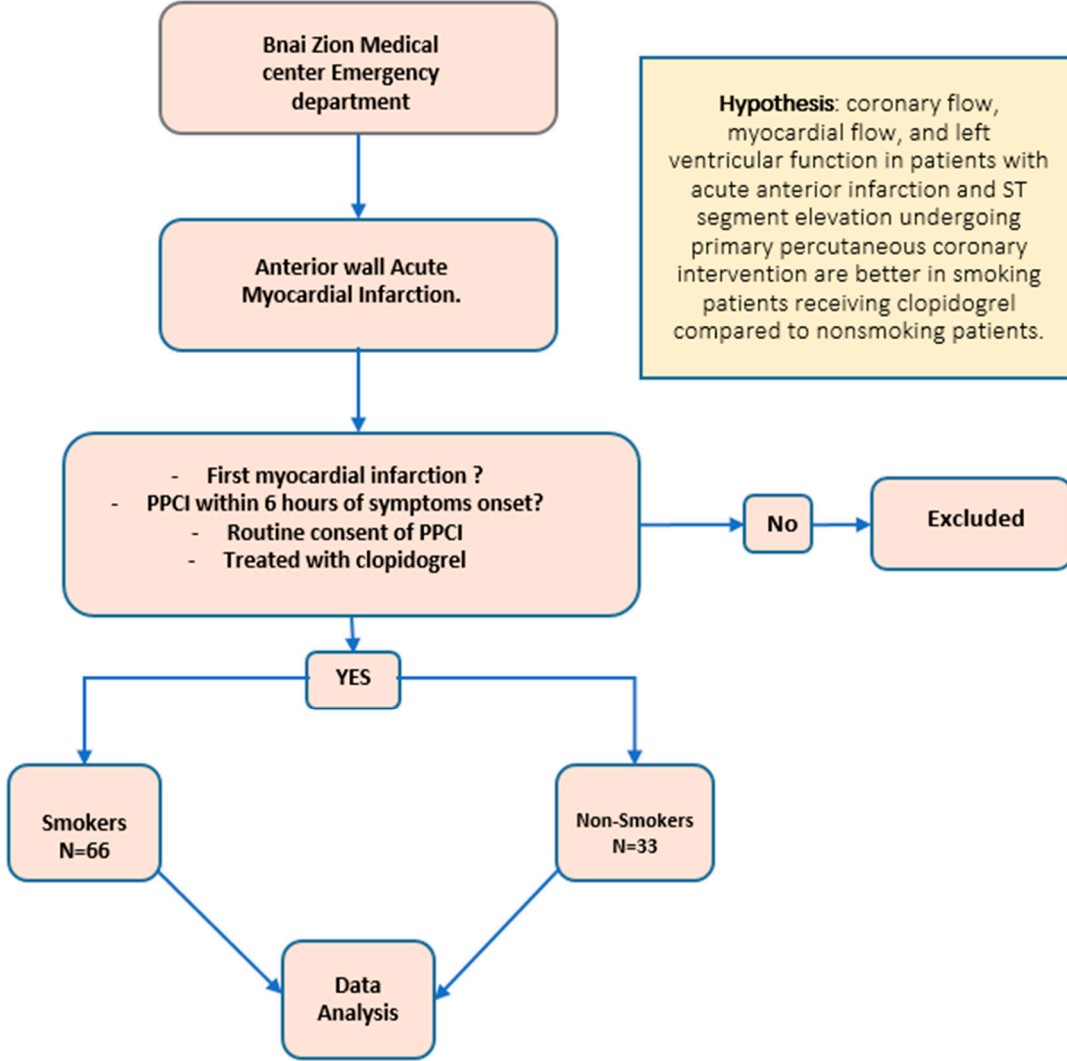

**Figure 1.** Flowchart of patient selection.

An electrocardiogram (ECG) was performed upon admission to the emergency department as well as in the cardiac intensive care unit. In addition, another ECG test was performed after the general intervention. All patients underwent an angiographic evaluation of the flow in the LAD according to the TIMI grade and the flow to the heart muscle according to the MBG, before and after the intervention. All patients underwent a complete transthoracic Doppler echocardiogram as usual on the day after the percutaneous coronary intervention and also on the fifth day of hospitalization before discharge from the hospital. In addition, all the patients underwent echo Doppler sampling of the blood flow velocity in the LAD.

### 2.1. Inclusion Criteria

Patients who arrived with a diagnosis of acute anterior STEMI* for the first time in their lives and underwent PPCI were treated with clopidogrel and met the following criteria:

(1)   Their first anterior wall ST segment elevation myocardial infarction (STEMI);
(2)   Patients who underwent PPCI within six hours of the onset of the infarction symptoms;
(3)   Routine informed consent to perform primary PCI.

*Anterior STEMI definition: chest pain that lasts at least 30 min, with ST segment elevations of at least 2.0 mm in two or more pericardial leads upon electrocardiography (ECG).

*2.2. Exclusion Criteria*

(1)     PPCI performed after more than six hours from the onset of the infarction symptoms;
(2)     Patients with previous anterior STEMI;
(3)     Patients whose initial PPCI failed;
(4)     Patients who underwent bypass surgery in the past;
(5)     Significant disease in the left main coronary artery;
(6)     Patients with insufficient Doppler echocardiography and without the necessary data for the study.

*2.3. Data Collection*

Data were collected from 99 patients. The data included basic information such as age, cardiovascular risk factors (obesity, smoking, excess blood lipids, type 2 diabetes, and hypertension), as well as the average times from the beginning of the pain until receiving the treatment (divided into three times): (a) from the beginning of the pain until hospital admission; (b) from hospital admission until the catheterization room, and (c) from the onset of pain until the catheterization room. In addition, we collected routine tests that were performed, including blood count, differential, coagulation functions, and markers such as troponin and CPK. Auxiliary tests that were performed were also collected, including an electrocardiograph (ECG) and echo heart of the patients before and after primary percutaneous coronary intervention.

ECG (electrocardiography): An electrocardiography recording was performed as described above before and after the initial percutaneous intervention. We measured the withdrawal between the two measurements and divided the withdrawal (in percentages) into three groups:

1.     First group: no resolution—less than 30%;
2.     Second group: partial resolution—30–70%;
3.     Third group: complete resolution—more than 70%.

Echocardiography: We used a Siemens device with the Acuson system, using a 3.5–7 MHz transducer. The echocardiographic measurements of LVEF were performed from a biplane apical view using the Simpson method. We performed two tests on all patients, one in the initial six hours after the initial percutaneous intervention and a second repeat test five days after the intervention. The calculations were made using the following equations:

- The LAD wall motion score index (LAD WMSI) was calculated using the following formula (9 segments):

$$LAD - WMSI = \frac{\sum score\ of\ 9\ segments}{9}$$

- LV wall motion score index (LV WMSI) was calculated using the following formula (16 segments):

$$LV - WMSI = \frac{\sum score\ of\ 9\ segments}{16}$$

After the calculations, a division was made according to the mobility of the ventricle. 1 = wall with normal movement, 2 = hypokinesia, and 3 = akinesia.

Angiographic analysis: Two cardiologists reviewed the coronary angiograms. An evaluation was given for the TIMI grades and also the myocardial blush grades before and at the end of the intervention.

Primary percutaneous intervention PCI: After processing of the data mentioned above and making a definite diagnosis, the patients were transferred to the catheterization room to perform a primary percutaneous intervention, during which, using high pressure implantation methods, bare metal stents were inserted. The evaluation of the myocardial blush was performed by taking pictures at the end of the procedure at an angle of 30 degrees

to the right above or 90 degrees to the side with a prolonged CINE. After the procedure, the patients were treated with a loading dose of clopidogrel 600 mg per day before the catheterization, followed by a maintenance dose of 75 mg per day. The Triton-TIMI38 study, which compared the results of the anticoagulant treatments (clopidogrel vs. prasugrel) in patients with STEMI who were treated by PPCI, showed that prasugrel reduces mortality and morbidity but increases the chance of bleeding. Therefore, the patients were treated with prasugrel with a loading dose of 60 mg before the PPCI, followed by a maintenance dose of 10 mg per day. However, in elderly patients, treatment with clopidogrel was continued. In addition, an intravenous bolus of heparin was given (50–70 Unit/kg) to achieve a coagulation time of 250 ms. However, according to the results of the HORIZON study, treatment with bivalirudin in an intravenous bolus of 0.75 mg/kg followed by an infusion at a dose of 1.75 mg/kg/h for four hours after the operation can be given.

Velocity of the LAD and measurements: To estimate the velocity of blood flow in the LAD, the color Doppler method was used and the Nick West limit was set at 17 cm/sec. To search for diastolic dye flow in the anterior interventricular groove, a parasternal transverse view was used while rotating the transducer in a clockwise direction, and to calculate diastolic blood flow velocity in the LAD, a shortened two-chamber apical view was used while rotating the transducer in a counterclockwise direction. To calculate an average of the LAD blood velocity curve parameters, an assessment was made of the area under the flow velocity curve, the diastolic deceleration time of velocity in the LAD was measured as the time from the peak diastolic velocity of the tangent of the velocity envelope to the baseline. Pressure half time (PHT) (ms) was determined as the time it takes for the peak diastole velocity to decrease to its initial $1/\sqrt{2}$ value.

## 2.4. Independent Variables

Smoking cigarettes or patients who did not smoke cigarettes.

## 2.5. Dependent Variables

Systolic function of the left ventricle: the ejection fraction in the left ventricle according to the Simpson method on admission and upon discharge of the patients from the hospital, the motility index of the left ventricle, and the motility index of the segments that receive blood supply from the LAD.

$$LAD - WMSI = \frac{\sum score\ of\ 9\ segments}{9}$$

$$LV - WMSI = \frac{\sum score\ of\ 9\ segments}{16}$$

where normal LV wall motion = 1, hypokinesis = 2, and akinesia = 3 for each of the segments.

Measures in the catheterization room: TIMI grade and MBG before and after PPCI.

EKG: degree of resolution of ST segment elevation less than 30%, between 30 and 70%, and more than 70%.

Diastolic flow velocity in LAD after PPCI:

Maximum diastolic and systolic flow velocity, area under the flow velocity curve, diastolic deceleration time of the velocity in the LAD. Evaluation of LAD blood flow velocity parameters immediately up to six hours after the intervention and later on the day of discharge.

## 2.6. Blood Viscosity

The viscosity is calculated according to the value of the hematocrit (Hct) and the value of the total protein in the blood (TP) through a formula that calculates the true viscosity of the blood (whole blood viscosity—WBV).

We used two formulas. One is used for high shear forces and the other for low forces. For high shear forces, WBV = (0.12 × Hct) + 0.17(TP − 2.07), and for low shear forces,

WBV = (1.89 × Hct) + 3.76(TP − 78.42). Both formulas are acceptable but not clinically valid. Despite this, it is still not known which of the two is better.

### 2.7. Statistical Methods

The values will be presented as mean and standard deviation. A comparison of the data between the different patient populations was made, and an ANOVA statistical calculation of the indices was performed. A multivariate analysis was performed. $p < 0.05$ was considered statistically significant.

The assessment of the strength of the study was made based on the expected difference in diastolic flow velocity in the control group, 31 ± 9 cm per second (mean ± SD), and a similar previous study group, 39 ± 11 cm per second (mean ± SD), with a $p$-value = 5% and the prevalence of a reversal of systolic flow as evidence of severe disturbance in microcirculation in the control group of 17% and, in the study group, 0%. Assuming that the significance is equal to 5% one way and there will be about 80 patients in the research group which is controlled against itself, the strength of the study will be sufficient, and about 90% that the difference between the groups is statistically significant. To estimate the expected values, we relied on previous experiments that appear in the literature.

### 2.8. Ethical Aspects

This research work received Helsinki approval (0001-09-BNZ) from the ethics committee of Bnai Zion Medical Center, as accepted. In addition, all the operations and tests performed on the patients were non-invasive, routine, without side effects, without radiation, acceptable, and performed with or without the current study for all patients with acute STEMI.

## 3. Results

Patient characteristics: 99 patients were examined, and the average age of the participants was 60.4. Significant differences were observed in the average age between the group of smokers and non-smokers, with an average of 54.7 versus 71.7, respectively. The number of women was 21 (21%), of which 36.3% were non-smokers and were almost three times the number of smokers (13.6%). In cardiovascular risk factors, larger percentages were seen in the non-smoker group, except in the category of family history of coronary disease where the values were 36.3% in smokers compared to 24.2% in non-smokers. In addition, the percentage of type 2 diabetes patients in both groups was almost the same.

Statistically significant differences (0.000003) of hemoglobin were observed in smokers (14.92 ± 1.03) compared to non-smokers (12.91 ± 1.95). We also observed statistical differences between the platelet levels in the smoking group (274.9 ± 69.27) compared to the non-smoking group (247.3 ± 54.77) with a statistical significance of 0.037. The levels of white blood cells in smokers were also higher (12.73 ± 4.02) compared to the non-smoker group (10.14 ± 2.63), $p = 0.0003$. Also, in reference to Killip, no differences were seen between the two groups (Table 1).

Statistically significant differences were also demonstrated between the viscosities of the group of smokers in the two calculation formulas. High shear values demonstrated a result of 6.14 ± 0.453 in smokers compared to 5.37 ± 0.77 in non-smokers with a statistically significant difference of 0.00005. Additionally, the calculation through the low shear formula demonstrated a viscosity of −185.64 ± 7.8 in the group of smokers compared to −197.64 ± 12.33 in the second group with a statistical significance of 0.00008.

Coronary flow parameters testing findings: Coronary flow parameters were tested before and after the initial percutaneous intervention as explained above. No statistically significant differences were observed in pre-TIMI between smokers and non-smokers (0.9 ± 1.2 and 0.84 ± 1.24, respectively, $p = 0.83$). In addition, no statistically significant improvement was observed post TIMI. Regarding MBG, here too we did not see significant statistical differences between smokers and non-smokers pre-MBG (0.3 ± 0.55 compared to 0.1 ± 0.31, respectively). Note that no one (0%) of the two groups had a pre-MBG value

of 3. After the intervention, the post-MBG results did not show statistical significance between the two groups, with values of 2.42 ± 0.6 in smokers compared to 2.92 ± 0.53 in non-smokers. The improvement in the percentage of people with post-MBG values of 3 was higher in people who smoke (37.8%) compared to people who do not smoke (24.2%) (Table 2).

**Table 1.** Patient characteristics and clinical findings.

| Characteristic — Anti-Coagulation PLUS | Smokers | Non-Smokers | *p*-Value |
|---|---|---|---|
| Number | 66 | 33 | - |
| Age (median) | 54.7846 ± 10.0195 | 71.78863 ± 10.0011 † | $3.7536 \times 10^{-11}$ † |
| Women (%) | 13.6% | 36.3% | - |
| Obesity (%) | 28 ± 45 | 35 ± 49 | 0.56 |
| Dm-type 2 (%) | 31 ± 47 | 24 ± 44 | 0.514 |
| HTN (%) | 55 ± 5 | 56 ± 5 | 0.957 |
| HPL (%) | 69 ± 47 | 68 ± 48 | 0.949 |
| PVD % | 9 ± 29 | 11 ± 32 | 0.839 |
| Previous CAD (%) | 13 ± 34 | 24 ± 44 | 0.298 |
| FH to CAD (%) | 45 ± 54 | 28 ± 46 | 0.174 |
| Killip 1 class (%) | 1.28 ± 0.74 | 1.16 ± 0.45 | 0.32 |
| Pain to door (min) | 98.7 ± 90.8 | 101 ± 96.9 | 0.92 |
| Door to balloon (min) | 127 ± 104 | 115 ± 108 | 0.6 |
| Pain to balloon (min) | 225 ± 138 | 216 ± 146 | 0.79 |
| Hb (gr/dL) | 14.92 ± 1.03 | 12.91 ± 1.95 | 0.000003 |
| Platelets ($10^3$/microliter) | 274.9 ± 69.27 | 247.3 ± 54.77 | 0.037 |
| WBC (mcL) | 12.73 ± 4.02 | 10.14 ± 2.63 | 0.0003 |
| High shear WBV (cP) | 6.14 ± 0.453 | 5.37 ± 0.77 | 0.00005 |
| Low shear WBV (cP) | −185.64 ± 7.8 | −197.64 ± 12.33 | 0.00008 |
| CRP (mg/L) | 10.36 ± 11.13 | 25.77 ± 48.68 | 0.166 |
| Aspirin | 100% | 100% | - |
| Bilavirudin | 100% | 100% | - |
| Clopidogrel | 62% | 72% | - |
| Prasugrel | 38% | 28% | - |

Dm-type 2—diabetes mellitus type 2; HTN—hypertension; HPL—hyperlipidemia; PVD—peripheral vascular disease; CAD—coronary artery disease; FH—familial history; pain to door—time in minutes since the onset of the pain until reaching the hospital; door to balloon—time in minutes since reaching the hospital until getting the treatment; pain to balloon—time in minutes since the onset of the pain until getting the treatment; WBV—whole blood viscosity; high shear—WBV = (0.12 × Hct) + 0.17(TP − 2.07); low shear—WBV = (1.89 × Hct) + 3.76(TP − 78.42). Hct (%), TP (g/dL). CRP—C-reactive protein. †—statistical significance.

Electrocardiographic findings: In half of the non-smokers, we saw a retraction of the ST segment of more than 70% before and after the intervention compared to a third of the smoking patients. Conversely, the percentage of non-smokers who had not experienced any withdrawal (less than 30%) was twice than that of smokers (16.6%) (Table 3).

Doppler flow data in the descending left coronary artery: In the early flow, we see equal values between smokers and non-smokers in diastolic filling time (11.9) compared to systolic filling time which is twice as slow in smokers (6.48 ± 16.4) compared to non-smokers (3.26 ± 1.53). When examining the other parameters of the early flow, no statistically significant changes were seen between smokers and non-smokers (Table 4A).

**Table 2.** Angiographic coronary flow parameters.

| Parameters | Smokers | Non-Smokers | *p*-Value |
|---|---|---|---|
| **Pre-TIMI** | $0.9 \pm 1.2$ | $0.84 \pm 1.24$ | 0.83 |
| **% TIMI = 3** | 13.6% | 18.1% | - |
| **Post TIMI** | $2.98 \pm 0.23$ | $2.44 \pm 0.77$ | 0.1 |
| **%TIMI = 3** | 83.3% | 81.8% | - |
| **Pre-MBG** | $0.3 \pm 0.55$ | $0.1 \pm 0.31$ | 0.06 |
| **%MBG$\leq$2** | 100% | 100% | - |
| **%MBG = 3** | 0% | 0% | - |
| **Post MBG** | $2.42 \pm 0.6$ | $2.92 \pm 0.53$ | 0.33 |
| **%MBG$\leq$2** | 62.12% | 75.7% | - |
| **%MBG = 3** | 37.8% | 24.2% | - |

Pre-TIMI: TIMI grade prior to PPCI; Post TIMI: TIMI grade after PPCI; Pre-MBG: myocardial blush grade (MBG) prior to PPCI; Post MBG: MBG after PPCI. %TIMI = 3: percentage of patients with TIMI grade flow "3"; %MBG $\leq$ 2: percentage of patients with MBG $\leq$ 2; %MBG = 3: percentage of patients with MBG = 3.

**Table 3.** ST-Segment elevation resolution.

| Resolution | Smokers | Non-Smokers | *p*-Value |
|---|---|---|---|
| **Complete > 70%** | 33.3% | 50% | - |
| **Partial (30–70)%** | 50% | 16.6% | - |
| **No < 30%** | 16.6% | 33.3% | - |
| **Average (mm)** | $1.6563 \pm 1.0119$ | $0.8 \pm 0.8367$ | 0.094 |

Doppler flow data in the left descending coronary artery: The late flow showed a difference of almost 0.5 mm in the diameter of the left descending coronary artery in favor of non-smokers. Statistically proven differences (*p*-Value = 0.01) were seen in diastolic flow velocity in non-smokers ($50 \pm 15.1$) vs. smokers ($38 \pm 13.6$). There were no statistically proven differences in systolic flow velocity between smokers and non-smokers, with values of ($16.6 \pm 13$) and ($20.2 \pm 8.58$), respectively.

In addition, statistically significant differences ($p = 0.05$) were observed in the diastolic filling time (TVI-D) between the non-smoker group ($15.5 \pm 4.4$) and the smoker group ($12.8 \pm 4.41$) (Table 4B).

Left ventricular systolic function: No statistically significant difference was demonstrated between left ventricular function between the group of smokers and non-smokers in the early and late examination. However, the difference between the two tests ($\Delta$EF) showed that the improvement in the smoking group ($6.183 \pm 4.37$) was twice as great as the non-smoking group ($3.787 \pm 2.678$). The examination of the other parameters showed almost no differences between the group of smokers and the group of non-smokers (Table 5).

Myocardial damage markers: Creatine phosphokinase values were higher in the non-smokers ($1772 \pm 1721$) compared to the smokers' group ($1569 \pm 1353$), but no statistical significance was seen (0.56). In contrast, the maximum troponin values were higher in the group of smokers ($53.1 \pm 51.6$) compared to ($40 \pm 43.5$) the non-smokers, without statistical significance (0.22). During the tests, only maximum values were recorded in the patients (Table 6).

Comparing smokers receiving prasugrel and smokers receiving clopidogrel, no statistical significance were observed on TVI (D) and VD parameters between both groups, with 0.38 and 0.07, respectively (Table 7).

**Table 4.** (**A**): LAD Doppler early parameters. (**B**): LAD Doppler late parameters.

| (A) | | | |
|---|---|---|---|
| **Parameters AVG** | **Smokers** | **Non-Smokers** | ***p*-Value** |
| LAD diam (mm) | 2.35 ± 0.35 | 2.5 ± 0.84 | 0.75 |
| VD (cm/s) | 44.7 ± 13.9 | 44.9 ± 18.6 | 0.98 |
| VS (cm/s) | 14.5 ± 13.9 | 18.8 ± 6.93 | 0.23 |
| TVI D (cm) | 11.9 ± 4.37 | 11.9 ± 5.11 | 0.97 |
| TVI S (cm) | 6.48 ± 16.4 | 3.26 ± 1.53 | 0.42 |
| PHT (ms) | 156 ± 88.6 | 139 ± 62.5 | 0.53 |
| DDT (ms) | 559 ± 319 | 461 ± 232 | 0.29 |
| % DDT > 600 ms | 12.12% | 18.18% | - |
| HR (per/min) | 79 ± 12 | 77.5 ± 14.2 | 0.72 |
| (B) | | | |
| **Parameters AVG** | **Smokers** | **Non-Smokers** | ***p*-Value** |
| LAD diam(mm) | 2.4 ± 0.3 | 2.85 ± 0.64 | 0.5 |
| VD (cm/s) | 38 ± 13.6 | 50 ± 15.1 † | 0.01 |
| VS (cm/s) | 16.6 ± 13 | 20.2 ± 8.58 | 0.26 |
| TVI D (cm) | 12.8 ± 4.41 | 15.5 ± 4.4 † | 0.05 |
| TVI S (cm) | 3.41 ± 2.35 | 4.12 ± 2.08 | 0.29 |
| PHT (ms) | 190 ± 73.3 | 161 ± 67.7 | 0.17 |
| DDT (ms) | 637 ± 239 | 561 ± 245 | 0.3 |
| % DDT > 600 ms | 28.7% | 18.8% | - |
| HR (per/min) | 68.5 ± 11.7 | 70.9 ± 10.9 | 0.52 |

LAD diam: left anterior descending diameter; VD: peak diastolic velocity; VS: peak systolic velocity; TVI D: diastolic time velocity integral; TVI S: systolic time velocity integral; PHT: pressure half time; DDT: diastolic deceleration time; HR: heart rate. †—statistical significance.

**Table 5.** Left ventricular systolic function.

| **Function Parameters** | **Smokers** | **Non-Smokers** | ***p*-Value** |
|---|---|---|---|
| **EF early (%)** | 37.3 ± 6.71 | 38.2 ± 6.02 | 0.54 |
| **EF late (%)** | 43 ± 9.49 | 42 ± 7.07 | 0.43 |
| **ΔEF** | 6.183 ± 4.37 | 3.787 ± 2.678 | |
| **LV WMSI early** | 1.6 ± 0.29 | 1.58 ± 0.22 | 0.75 |
| **LV WMSI late** | 1.49 ± 0.28 | 1.5 ± 0.24 | 0.92 |
| **ΔLV WMSI** | −0.109 ± 0.077 | −0.0775 ± 0.005 | |
| **LAD WMSI early** | 2.0064 ± 0.490 | 2.0265 ± 0.444 | 0.889 |
| **LAD WMSI late** | 1.7909 ± 0.4673 | 1.7766 ± 0.4752 | 0.92 |
| **ΔLAD WMSI** | −0.215 ± 0.152 | −0.249 ± 0.176 | |

EF early: early left ventricular ejection fraction; EF late: late left ventricular ejection fraction; ΔEF: delta of EF early and EF late; LV WMSI early: early left ventricular wall motion index; LV WMSI late: late left ventricular wall motion index; ΔLV WMSI: delta of LV WMSI late and early; LAD WMSI early: early wall motion score index of segments supplied by the left anterior descending coronary artery; LAD WMSI late: late wall motion score index of segments supplied by the left anterior descending coronary artery; ΔLAD WMSI: delta of LAD WMSI early and late.

**Table 6.** Biomarker maximal blood levels.

| Biomarkers | Smokers | Non-Smokers | *p*-Value |
|---|---|---|---|
| Max CPK | 1569 ± 1353 | 1772 ± 1721 | 0.56 |
| Max troponin | 53.1 ± 51.6 | 40 ± 43.5 | 0.22 |

Max CPK: maximal creatine phosphokinase.

**Table 7.** Comparison of drug effects of smoker group.

| | Clopidogrel. | Prasugrel | *p*-Value |
|---|---|---|---|
| **TVI D(cm)** | 13.38 ± 16.85 | 11.94 ± 23.99 | 0.38 |
| **VD (cm/s)** | 40.77 ± 199 | 52.27 ± 315 | 0.07 |

TVI D: diastolic time velocity integral, VD: peak diastolic velocity.

## 4. Discussion

Reperfusion treatments contributed in a very significant way to reducing mortality as a result of STEMI. In addition, percutaneous intervention and antithrombolytic treatments also contributed to improving survival from the event. Smoking is known as a significant risk factor for coronary disease formation, but paradoxically, despite the higher incidence of STEMI infarcts in patients who smoke, previous studies have shown that mortality after STEMI is lower in smokers compared to non-smokers, especially after fibrinolytic therapy. This phenomenon has been defined as "The Smoker's Paradox" [28–33].

In the present study, 99 patients with acute myocardial infarction who underwent primary percutaneous coronary intervention were examined. Two-thirds of the patients, 66 in number, were cigarette smokers and were mostly young men (the groups had a lower frequency of women). In the smoking patients, a reduced incidence of the complete retraction of ST segment elevation was found compared to the non-smoking patients. It was also found that the diastolic flow integral and its velocity in the LAD were significantly lower in the group of smokers compared to non-smokers.

In contrast, in the angiographic parameters, the level of myocardial markers in the blood and the systolic function of the left ventricle at admission and discharge were not affected by smoking and were similar in both groups. Therefore, the research hypothesis that coronary flow, myocardial flow, and left ventricular function in patients with acute anterior infarction and ST segment elevation who undergo primary percutaneous coronary intervention were better in smoking patients who received clopidogrel was proven incorrect. Rather, some of the data, as mentioned above, were found to be better in the non-smokers, and the rest of the data were similar between the smokers and the non-smokers.

Conventional risk factors for STEMI include male gender, smoking, and a family history of early CAD [28]. According to this study, we proved with statistical significance ($3.7536 \times 10^{-11}$) that STEMI in the anterior wall appeared at a younger age in people who smoke ($54.8 \pm 10.0$) compared to people who do not smoke ($71.9 \pm 10.0$). In addition, in our study, the current smokers were not only younger but there was also a greater prevalence of men who also tend to have a lower prevalence of other risk factors such as high blood pressure, excess lipids in the blood, obesity, and previous coronary disease.

In a previous prospective study that examined 1068 patients with STEMI and compared groups of smokers and non-smokers, similar angiographic values were found without statistical significance ($p = 0.62$) of the percentage of smokers with TIMI grade 2–3 (28.9%) compared to the percentage of TIMI grade = 2–3 in non-smoking patients (27.5%). After PPCI was performed in both groups, TIMI = 3 rates were found to be higher in the smoking group (94.1) compared to the non-smoking group (89.6) with a statistical significance of 0.016 [29]. Similarly, in our study, after checking the angiographic parameters detailed above, we found no statistical differences between the TIMI values between the two groups, but there was a better improvement in the values in the group of smokers (600%) compared to the group of non-smokers (450%) without statistically significant differences ($p = 0.1$).

In another study that examined whether time until primary percutaneous intervention affects ECG resolutions and the MBG, 1072 patients with STEMI who were treated with primary percutaneous intervention were examined, and myocardial reperfusion was assessed by the ST elevation resolution test and the MBG. The time taken until the treatment (time-to-treatment) was in direct correlation with the size of ST elevation resolution (adjusted OR [95% CI] ¼ 1.01 [1.01–1.02], $p < 0.001$) and the MBG (adjusted OR [95% CI] ¼ 1.01 [1.01–1.02], $p < 0.0001$) [6]. In our study, and in contrast to the previous study, when testing the MBG levels between the group of smokers and non-smokers, no statistically significant differences were found, but the percentage of smokers with MBG = 3 was higher than the percentage of non-smokers (37.8% compared to 24.2%, respectively). During the study, we divided the patients according to the retraction of the ST segment into 3 groups: up to 30%, between 30 and 70%, and over 70. No statistical significance was demonstrated ($p = 0.094$) in the average change between the group of smokers (1.0119 ± 1.6563) compared to the non-smoker group (0.8 ± 0.8367).

From a pathophysiological point of view, cigarette smoking continues to be a major health risk, which contributes significantly to cardiovascular morbidity and mortality [30]. Cigarettes accelerate arteriosclerosis, first through the vasomotor properties of the arteries which impair their ability to expand. This has been attributed to the action of nitric oxide (NO) whose availability is impaired as a result of smoking cigarettes. Furthermore, inflammation starts a cascade of events such as the recruitment of white blood cells that start the chain of the production of plaques, in addition to the poor functioning of the endothelium up to acute clinical events, which is manifested by an increase in markers such as C-reactive protein (CRP) and interleukin. It was also found that passive smoking is also a significant factor in the development of cardiovascular risks, and therefore, in our opinion, cardiac ischemia appears at an earlier age in patients who smoke compared to non-smokers, which is statistically proven in this work [31–36].

In another study, the viscosity was tested in 74 smoking patients, and it was found that a less preferred viscosity is found in the smokers compared to the non-smokers. The study also found that stopping smoking for three months resulted in a reduction in the viscosity and its effects on cardiac blood flow. These data are consistent with the results of the current study, since the viscosity among the smokers was greater than the non-smokers group in a statistically significant way. It should be noted that this result is valid for both calculation formulas for blood viscosity [37–41].

In a previous study that examined 31 patients who presented with anterior MI, flow parameters were examined after six hours, 36–48 h, and five days after a successful primary percutaneous intervention. The tests showed that within two days of the percutaneous intervention, DDT values (600 ± 340 ms) were shorter than the intervention on the fifth day (807 ± 332 ms) ($p < 0.012$). In our study, the percentage of people with a diastolic deceleration time >600 (%DDT > 600) was unchanged before and after the intervention in non-smokers (18.8%) compared to an improvement in the group of smokers in the early flow (12.12%) compared to the result (28.7%) in the later test (Table 4B).

As for the paradox the research raises, smoker patients were significantly younger than the non-smoker group (54 years and 71 years, respectively). Our results are consistent with a Malaysian report of 12,400 smokers vs. 10,666 non-smokers concluding with the same paradox, and it was attributed to younger smoker age compared to non-smokers and lower cardiovascular burden. Consistently, our smoker group had a lower prevalence of obesity, hypertension, and peripheral vascular disease. One other explanation relies on the shear viscosity which was significantly different between the groups, with a higher mean for the smokers group compared to non-smokers (6.14 ± 0.453 and 5.37 ± 0.77, respectively). Blood viscosity depends on its shear rate: at a higher shear, blood cells disaggregate and align in the direction of flow; in contrast, at a lower shear, blood cells aggregate with an increase in the viscosity and therefore might appear in sites of atherosclerotic obstruction and limited flow, playing a critical role [42]. Lower shear jeopardizes the tissue perfusion by higher resistance rates. Furthermore, our study found significant differences in hemoglobin levels,

with 14.92 ± 1.03 and 12.91 ± 1.95 in smokers and non-smokers, respectively. Anemia was correlated to a lower rate of complete revascularization compared with normal values [43], which again may explain the paradox in our outcomes.

This study has several limitations. We investigated 99 patients who were hospitalized in our medical institution, and therefore, this is a small number of subjects in one research center. Our results and hypotheses should therefore be confirmed in large multicenter studies. In addition, the smoking status was assessed only at the time of admission to the hospital based on the patient's admission. Accurate information about the patient's smoking history (smoker pack years, quitting year) was not recorded in the medical file, so this could affect the results of the "non-smoker" group. Nonetheless all patients did not smoke during hospitalization.

Despite these limitations, the research reveals that smoking patients had a lower incidence of complete ST elevation retraction compared to non-smokers, and the diastolic flow integral and its velocity in LAD were lower in the smoker group.

## 5. Conclusions

Cigarette smokers with anterior STEMI treated via PPCI were younger with a lower prevalence of women and of complete ST elevation resolution and had lower LAD diastolic velocity and integral late after PPCI. However, angiographic parameters and LV systolic function parameters were similar.

**Author Contributions:** M.B. and D.S. contributed to the design, screening, and data extraction and analysis and wrote this review. M.B., Y.S. and D.S. contributed to design, data analysis, and interpretation and critically reviewed the paper. M.B., N.A., Y.S. and D.S. contributed to the design and search strategy and performed the literature search. M.B., Y.S., N.A. and D.S. contributed to the screening and data analysis. M.B. and D.S. conceived the project, contributed to the design, review, data analysis and interpretation, resolved conflicts, and critically reviewed the paper. All authors have read and agreed to the published version of the manuscript.

**Funding:** This study did not receive any specific grants from funding agencies in the public, commercial, or not-for-profit sector.

**Institutional Review Board Statement:** This research work received Helsinki approval (0001-09-BNZ) from the ethics committee of Bnai Zion Medical Center, as accepted.

**Informed Consent Statement:** Patient consent was waived due to retrospective approach of the study.

**Data Availability Statement:** The data presented in this study are available on request from the corresponding author. The data are not publicly available due to privacy of the patients.

**Acknowledgments:** We wish to thank Ola Boulos for her critical role in supporting us through the writing and editing of our paper and extracting the Helsinki approval. We would to thank the directors and medical teams of the Internal Medicine Departments in Galilee Medical Center, Nahariya, Israel, for their help and support.

**Conflicts of Interest:** The authors declare no conflicts of interest.

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
