# Peer review of "Significance of Smoking in Patients with Acute ST Elevation Myocardial Infarction (STEMI) Undergoing Primary Percutaneous Coronary Intervention: Evaluation of Coronary Flow, Microcirculation and Left Ventricular Systolic Function"

_hearts, doi:10.3390/hearts5010012_

Round 1

Reviewer 1 Report

Comments and Suggestions for Authors

Thank you very much for inviting me to review your work:

"Significance of Smoking in Patients with Acute ST Elevation Myocardial Infarction (STEMI) Undergoing Primary Percutaneous Coronary Intervention: Evaluation of Coronary Flow, Microcirculation and Left Ventricular Systolic Function"

The work raises a very important issue of the involvement of classic cardiovascular risk factors in STEMI. The manuscript is written correctly. I do not see any significant substantive errors.

Minor comments: The introduction is written correctly. discusses important coronary risk factors. To increase the didactic value of the work, I suggest including a graphic abstract showing classic and non-classical coronary risk factors. I propose to quote a paper discussing classic and less classic risk factors: PMID: 37971709.  The research hypothesis is well formulated.

Material and methods: I suggest including a flowchart for an even better graphical representation of the study group.

Results: the results are well described and contain 6 very large tables.

Author Response

thank you very much for your review. we added the following flow chart and Graphical Abstract as recommended. ( attached in the file) 

Reviewer 2 Report

Comments and Suggestions for Authors

The manuscript entitled Significance of Smoking in Patients with Acute ST Elevation Myocardial Infarction (STEMI) Undergoing Primary Percutaneous Coronary Intervention: Evaluation of Coronary Flow, Microcirculation and Left Ventricular Systolic Function is an original paper.

The authors assessed retrospectively if coronary flow, myocardial flow, and left ventricular function in patients with acute anterior infarction and ST segment elevation undergoing primary percutaneous coronary intervention are better in smoking patients receiving Clopidogrel compared to nonsmoking patients.

Major revision

The authors concluded that smoking patients had lower incidence of complete ST elevation retraction compared to non-smokers. How do you explain that? Please explain clearly this ``paradox``. What were the other factors, which might concurred to this?

The authors noted that smoking increases the availability of the active metabolite of Clopidogrel which increases its activity in smokers. Another major limitation of this study is the fact that the drugs were not specified. Please add all pharmacologic treatment in both groups.

In study limitations the authors said: ``Accurate information about the patient’s smoking history was not recorded in the medical file, so this could affect the results of the "nonsmoker" group.`` This is a major limitation and the main cornerstone of this study. Do they continued smoking during the hospitalization?

If do you not have strong explanations for the above comments, this study has unacceptable issues.

Minor revision

 Mean value is recommended to be noted as x± SD (standard deviation)

Author Response

thank you for your comments. we addressed it with the attached file ! 

Round 2

Reviewer 2 Report

Comments and Suggestions for Authors

Thank you for responding and changing the manuscript.